# Evaluation of Prolonged Walking in Persons with Multiple Sclerosis: Reliability of the Spatio-Temporal Walking Variables during the 6-Minute Walk Test

**DOI:** 10.3390/s21093075

**Published:** 2021-04-28

**Authors:** Nawale Hadouiri, Elisabeth Monnet, Arnaud Gouelle, Pierre Decavel, Yoshimasa Sagawa

**Affiliations:** 1Laboratory of Clinical Functional Exploration of Movement, University Hospital of Besançon, F-25000 Besançon, France; Pierre.Decavel@h-fr.ch (P.D.); ysagawajunior@chu-besancon.fr (Y.S.); 2Clinical Investigation Center, University Hospital of Besançon, INSERM 1431, F-25000 Besançon, France; elisabeth.monnet@univ-fcomte.fr; 3Department of Physical Medicine and Rehabilitation, Dijon-Bourgogne University Hospital, F-21000 Dijon, France; 4Gait and Balance Academy, ProtoKinetics, Havertown, PA 19083, USA; arnaud.gouelle@gmail.com; 5Laboratory Performance, Santé, Métrologie, Société (EA7507), UFR STAPS, F-51100 Reims, France; 6Integrative and Clinical Neurosciences EA481, Bourgogne Franche-Comte University, F-25000 Besançon, France; 7Rehabilitation Department, HFR, CH-1700 Fribourg, Switzerland

**Keywords:** 6MWT, walking, multiple sclerosis, reliability, perceived exertion

## Abstract

Background: Walking disorders represent the most disabling condition in persons with Multiple Sclerosis (PwMS). Several studies showed good reliability of the 6-min walk test (6MWT) (i.e., especially distance traveled), but little is known about the reliability of the Spatio-temporal (ST) variables in the 6MWT. Objective: To evaluate the test-retest reliability of ST variables and perceived exertion during the 6MWT in PwMS and comparable healthy persons. Methods: We explored three 1-min intervals (initial: 0′–1′, middle: 2′30″–3′30″, end: 5′–6′) of the 6MWT. Six ST variables and perceived exertion were measured (respectively, using the GAITRite system and the Borg Scale). These measurements were performed twice, 1 week apart. The test-retest effects were assessed using the intraclass correlation coefficient (ICC) or the weighted kappa. Results: Forty-five PwMS and 24 healthy persons were included. The test-retest reliability of ST variables values was good-to-excellent for PwMS (ICC range: 0.858–0.919) and moderate-to-excellent for healthy persons (ICC range: 0.569–0.946). The test-retest reliability values of perceived exertion were fair for PwMS (weighted kappa range: 0.279–0.376) and substantial for healthy persons (weighted kappa range: 0.734–0.788). Conclusion: The measurement of ST variables during these 6MWT intervals is reliable and applicable in clinical practice and research to adapt rehabilitation care in PwMS.

## 1. Introduction

Multiple sclerosis (MS) is a progressive chronic inflammatory disease of the central nervous system. It is the main neurological cause of functional disability in young adults (i.e., between 20 and 49 years old) [1,2]. It results in multiple neurological deficiencies—sensorimotor, balance, cognitive deficits, and fatigue—with a major impact on walking in persons with MS (PwMS) [3]. Moreover, in more than two-thirds of PwMS, a walking disorder is the most reported limiting activity [4] and contributes negatively towards PwMS’s participation in their daily activities [4].

Understanding the severity of walking disorders in PwMS is essential for providing optimal medical care [5]. Although the Expanded Disability Severity Scale (EDSS) has been considered the gold standard for diagnosing the clinical and functional severity of MS, recent studies have highlighted its limitations and suggested that the EDSS should be supplemented by other objective standardized measurements [6]. The main limitation of the EDSS is that the walking capacity is self-reported by PwMS [7]; this subjective assessment with a possible reporting bias delivers information that could be influenced by cognitive deficits and desire to impress the healthcare provider [8].

Different tests have been used to assess the walking disorders noted in MS. Collectively, walking tests can be classified as short or prolonged tests [8]. The well-known short tests are the timed 25-foot walk (T25FW) [9] and the 10-m test [10]. These tests have the advantage of being simple to organize and apply; however, they have lower ecological properties compared to prolonged tests [11]. The most commonly recommended prolonged tests are the 2-Minute Walk Test (2WMT) and the 6-Minute Walk Test (6MWT) [8]. They highlight motor fatigue resulting from extended task execution, thus, effectively assessing the physical efforts and level of autonomy of PwMS [12,13,14]. For these reasons, the prolonged tests are a better indicator of the ability of PwMS to perform the activities of daily living and have been evaluated for use in clinical practice and research [8,15,16]. Studies in PwMS [15,17,18,19] and in other populations [20,21] have demonstrated that some walking variables measured during a 6MWT like walking velocity, cadence, or stride time could be altered across the 6MWT. These results suggest that the measurement of variables other than the classic distance to complete the 6MWT could provide a more detailed idea of the patients’ performance. Spatio-temporal (ST) walking variables (e.g., speed, cadence, stride length, stride width, etc.) have been used to objectively assess walking disorders in short distance tests [22] with good reliability [23] in PwMS. Measurement of these variables in prolonged walking tests could improve our understanding of how PwMS manage their efforts during the 6MWT. Furthermore, studying the ST variables within specific intervals in PwMS could open up new opportunities for exploration during the 6MWT, as has been done in other diseases [20,21]. In one study on persons after a free vascularized flap fibula for auto reconstruction [20], the strategic intervals of the 6MWT (i.e., the initial, middle, and end of the 6MWT) seemed better than just the classic 6MWT distance to explore and distinguish adaptations and tolerance involved in performing a 6MWT [20]. The observed variations across and between these time intervals could improve our understanding of different patterns of physical exertion during the course of MS, thus, allowing for the adaption of interventional targets. In a complementary way to the ST variable measures, it is important to quantify the perceived exertion, which informs on the tolerance of the effort in PwMS across the 6MWT. For example, the Borg scale was used in several studies to evaluate the perceived exertion and fatigue during aerobic exercises, such as the 6MWT in PwMS [24,25,26]. In one study, there was a significant correlation between the distance traveled in the 6MWT and the Borg score self-reported at the end of the 6MWT [26].

Although several studies have highlighted good reliability of the 6MWT score (i.e., the distance traveled in meters measured only one time at the end of the 6MWT) [16], little is known about the reliability of the ST variables across the 6MWT in PwMS with several measures during the 6MWT [27]. Moreover, to our knowledge, no study has analyzed the reliability of perceived exertion during the 6MWT (e.g., using the Borg scale). ST parameters and perceived exertion analysis throughout the 6MWT could highlight an impoverishment of motor possibilities and adaptability due to fatigue in MS, leading to a more constrained pattern. Therefore, before considering the clinical application of the aforementioned particular approach, it is important to check the metric properties of the assessments used. The aim of this observational study was to evaluate the reliability of the ST variables and perceived exertion (i.e., the physical sensations a person experiences during physical activity) during specific intervals of the 6MWT in PwMS and comparable healthy persons.

## 2. Materials and Methods

### 2.1. Participants

The participants were recruited from the FAMPISEP project (NCT02849782). Briefly, the FAMPISEP project is a single-center Phase 4 trial that was conducted between April 2014 and March 2019 at the University Hospital of Besançon (France) to investigate the effects of fampridine (4-aminopyridine) in PwMS with walking troubles. However, in this ancillary study, fampridine action was not evaluated. Walking assessments were done before the fampridine intervention.

The eligible PwMS in the FAMPISEP project had (i) a diagnosis of MS according to the modified McDonald criteria [28]; (ii) the ability to walk for a period of at least 6 min; and (iii) an EDSS score of 4.0–6.5. We included PwMS in this range because it is between the values of 4.0 and 6.5 that gait worsening is rated. The exclusion criteria were: (i) worsening MS symptoms during the previous 60 days, (ii) medical treatment or rehabilitation protocol change in the previous 60 days and during the study period, and (iii) contraindications to fampridine (history of seizures or renal insufficiency with creatinine clearance less than 80 mL/min).

In this study, a group of healthy volunteers (healthy group) was recruited from the general community to make a comparable group with PwMS in terms of distribution of sex, age, body mass, body height, and body mass index (BMI); they presented no neuro-orthopedic problems or other antecedents that could compromise their walking capacity. Recruiting a healthy group may permit a better study of the impact of the disease in the PwMS group concerning the reliability of ST variables and perceived exertion during their 6MWT. Written informed consent was obtained from all of the participants included in this study. This protocol was governed by French legislation on interventional biomedical research and was submitted to the national ethics committee (#13/405). The study was approved by the French Health Products Safety Agency (#2013-A002305-56).

### 2.2. Procedures and Instrumentation

A practitioner screened all the participants for the inclusion and exclusion criteria and collected the basic clinical and demographic data (i.e., sex, age, body mass, height, BMI, and tobacco and alcohol exposition (e.g., any tobacco/alcohol regular use)). A neurological examination was performed for PwMS at baseline to determine the EDSS score, disease duration, and MS phenotype (i.e., relapsing-remitting (RR), secondary progressive (SP), or primary progressive (PP)). The walking test was then performed by a single evaluator in a dedicated room at a controlled temperature of approximately 22 °C. The 6MWT was performed in accordance with the instructions of the American Thoracic Society, with the exception that the participants performed the test on a 24-m circuit rather than the recommended 32-m circuit. The instructions of the test were read by the same examinator to the participants before starting the test, and they were encouraged every minute during the test. A GAITRite walkway system (6.10 m × 0.61 m) (CIR Systems Inc., Franklin, NJ, USA) was embedded in the circuit (Figure 1) to capture foot contacts. At the same time, the software PKMAS (ProtoKinetics, Havertown, PA, USA) was used to process and export the following ST variables for each walkway passage (Appendix A): velocity (m/s), cadence (steps/min), stride length (m), stride width (m), and double support time (% of gait cycle). These measurements were recorded in the following 1-min intervals during the 6MWT: initial, 0′–1′; middle, 2′30″–3′30″; end, 5′–6′ (Figure 1).

These intervals were chosen to better explore and distinguish the different strategical periods of adaptation and tolerance involved in performing a 6MWT, as we have reported previously [20]. Moreover, the walking assessments in these specific intervals respect the number of steps required to study the aforementioned ST variables [29]. Only the values recorded on the walkway in these time intervals were analyzed. Therefore, participants had to perform at least one pass on the GAITRite walkway per interval (i.e., initial, middle, and end) during the 6MWT to be included in the analysis. Perceived exertion was measured at the end of each interval by using the Borg Scale [30]. Finally, the total distance walked during the 6MWT was measured. This walking test was performed twice, one week apart. The one-week duration was chosen to minimize the learning effect while avoiding changes in performance due to the progression of MS. We ensured that identical procedures were performed between the test and retest.

### 2.3. Statistical Analysis

The statistical analyses were performed using SAS version 9.4 (SAS Inc., Cary, NC, USA). Assuming a minimal ICC of 0.5 against a desired of 0.8 based on α = 0.05 and β = 0.2, the minimal number of observations required in this study was 22 participants per group [31].

According to the nature of the variables, they were presented as mean and standard deviation (SD) or median and/or (1st–3rd quartiles) frequency distribution. For each participant and each ST measure, the quantitative variables were averaged for each 6MWT interval. We evaluated the normality of continuous variables distribution using the Kolmogorov–Smirnov test and the homogeneity of variance using Levene’s test.

Subsequently, the test-retest reliability was assessed for continuous variables using the intraclass correlation coefficient (ICC) with a 2-way random effects analysis of variance model. ICCs for each ST variable and their 95% confidence intervals (95%CI) were calculated in both groups for the 3 intervals of the 6MWT. ICC values were interpreted according to the Guidelines for Reporting Reliability and Agreement Studies (GRRAS) [32]; ICCs between 0.50 and 0.70 were considered as moderate, higher than 0.70 as good, and higher than 0.90 as excellent (i.e., to be applied in clinical practice). For the Borg values, as it is an ordinal variable, the test-retest reliability was assessed using weighted Kappa and a 95% CI in both groups for the 3 intervals of the 6MWT. Weighted Kappa values were interpreted according to Landis and Koch’s benchmarks; Kappa values between 0–0.2 were considered as slight, 0.21–0.40 as fair, 0.41–0.60 as moderate, 0.61–0.80 as substantial, and 0.81–1 as almost perfect.

Finally, Bland–Altman plots with 95% Limits of Agreement (LoA) were produced for continuous variables over each interval in both groups. These plots enable visualization of the agreement between the test and retest measurements to check for systematic bias and possible outliers. Bland–Altman plots’ homoscedasticity was verified using the Spearman’s rank correlation coefficient tested at *p* ≤ 0.05 for the means and absolute differences between the two measurements. If the correlation was significant (heteroscedasticity), logarithmic transformation of the variable was recommended [33].

## 3. Results

### 3.1. Characteristics of the Participants at Baseline

Overall, 45 PwMS and 24 healthy persons were included in this study, respecting the minimal number of participants per group (i.e., 22, see the Statistical analysis section). The baseline characteristics of these participants are summarized in Table 1. In the PwMS group, the median EDSS (1st–3rd quartiles) and the mean (SD) disease duration were 4.5 [4,5] and 17.4 (8.9) years, respectively. The major MS phenotype was SP (44.4%).

### 3.2. Test-Retest Reliability

Table 2 and Table 3 summarize the ICC values obtained for all walking variables, in the three intervals of the 6MWT and for both groups. The ICC values for the 6MWT distance were *good* (ICC (95%CI): 0.891 (0.791–0.942)) in the PwMS group and in the healthy group (ICC: 0.728 (0.350–0.846)).

The ICC values for the different ST variables were *good**-to**-excellent* in each 6MWT interval in the PwMS group (ICC ranged from 0.846 (0.696–0.929) to 0.919 (0.840–0.957)). In the healthy group, the ICC values were *moderate**-to**-excellent* (ICC ranged from 0.569 (0.201–0.791) to 0.946 (0.880–0.976)).

The Kappa values of perceived exertion for the PwMS group were *fair* (weighted Kappa ranged from 0.279 (0.105–0.454) to 0.376 (0.208–0.544)). For the healthy group, they were *substantial* (weighted Kappa ranged from 0.734 (0.543–0.925) to 0.788 (0.646–0.898)).

To illustrate the clinical significance of exploring the reliability of these 6MWT intervals, a graphical representation of the averaged walking velocities (i.e., an average of the test and retest values) of each interval of the 6MWT in both groups is illustrated in Figure 2. The patterns of evolution during the 6MWT were different between the groups. PwMS had a “constant decline” pattern (i.e., a monotonous decrease in velocity was observed during the 6MWT), whereas the healthy group had a “rebound” pattern (i.e., the initial velocity was higher than that of the middle interval and then, an increase in the velocity towards the end was noted, which almost reached the initial velocity).

## 4. Discussion

In our study, ST measures in PwMS demonstrated good-to-excellent test-retest reliability in each interval. Bland–Altman plots revealed excellent agreements between the tests for all variables with no systematic bias. However, the reliability of all perceived exertion based on Borg values was poor during each interval. A better understanding of how PwMS deal with prolonged exercises could be relevant in improving rehabilitation care (e.g., to determine PwMS-specific physical activity goals). Although the clinical pertinence, to the best of our knowledge, no study has previously determined the reliability of this assessment.

The test-retest reliability values of the 6MWT distance in PwMS were close to the previously reported values [15,23], suggesting the representativeness of our evaluation. The reliability of ST variables was better in PwMS than that observed in the healthy group. One of the main explanations of this result is that PwMS in this study presented severe MS according to their EDSS score. The median EDSS score in the present PwMS group was 4.5, corresponding to a maximum expected walking distance—according to the EDSS scale—of 300–500 m with no restriction of time. In our study, the mean distance in the 6MWT in the PwMS group was 349 m and 368 m during the test and retest, respectively. Considering the distances stipulated by the EDSS and those observed in this study, PwMS were able in 6 min to cover 90% of the distance that they can best do without time constraint [34]. The better reliability of ST variables in the present study could be explained by the fact that a 6MWT appears to be considered as a maximal stress test by PwMS. This hypothesis could be, in part, confirmed by another study in PwMS where there was better reliability in walking velocity in T25FW in fast conditions than at comfortable conditions [23]. Moreover, another study highlighted excellent reliability during a 6MWT (without the specific instruction to walk as long-distance as possible) with several walking parameter values acquired with small inertial sensors. In this study, the higher the EDSS score, the more the reliability was increased [27], which may explain the good to excellent reliability values in our study in persons with severe MS.

This study demonstrated fair reliability values for perceived exertion in the PwMS group during the 6MWT, whereas these reliability values in the healthy group were considered too good. These results suggest that when repeating an almost identical physical test (see 6MWT distance and ICC values), PwMS perceived their effort differently through the trials. Several factors, such as fatigability, depression, and impaired sensory feedback, can influence the perceived exertion in PwMS [3]. These results corroborate with previous studies that have demonstrated perceived exertion disorders in PwMS and the importance of exploring their walking disabilities by standardized and objective measures [6,12]. These aspects should be considered in futures studies.

One main limitation in our study was that the measurement of ST variables during the 6MWT was not continuous and was dependent on the length of the GAITRite instrumented walkway available (i.e., 6.10 m). Although this setup was the most feasible (i.e., no intervention in patients), only 25% of the total walkway was explored to measure the ST parameters during these tests. This technical issue could be contoured by using longer walkways, inertial sensors, or instrumented insoles that allow for ST variables to be acquired cycle-by-cycle over the course of the circuit [35,36], permitting to evaluate PwMS who walk slower and have a more severe form of MS. A recent study in PwMS, only with an SP phenotype, explored this solution with small inertial sensors during a 6MWT with excellent test-retest reliability values for several ST parameters [27]. In our study, only ST variables were used to assess walking. For example, in PwMS, spasticity could be influenced by the prolonged effort [37] and result in changes in joint displacements and movements [38]. It might be interesting in the same context to implement other evaluations for measuring physical exertion management, like joint kinematics, kinetics, or muscle activation [39]. Finally, only two tests were performed in this study. For example, to achieve acceptable walking reliability during a 6MWT in healthy persons, one study proposed two practice rounds of the 6MWT before the actual tests [40]. However, a greater number of evaluations within a short period of time could reduce the feasibility of the evaluation, especially in clinical practice. Moreover, increasing the number of evaluations may also limit the interpretations of this study because of the potential for induced fatigue in PwMS.

## 5. Conclusions

Our results demonstrate that ST variables analyzed during specific 6MWT intervals are reliable for assessing the physical management in PwMS. This approach could be used in both clinical practice and research to evaluate the advantages of physical or medical treatments during the specific intervals of a 6MWT. Perceived exertion assessment by the Borg scale during a 6MWT was only reliable in the healthy group, suggesting that for the same physical effort, PwMS have difficulty in perceiving the same exertion. These results highlight that, as much as possible, walking assessments of PwMS should be done by objective measurements.

## Figures and Tables

**Figure 1 sensors-21-03075-f001:**
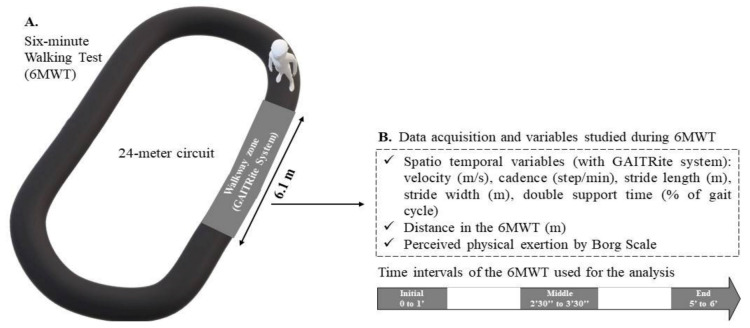
(**A**) Evaluation of gait during the 6-min walk test (6MWT); (**B**) variables studied during each interval of the 6MWT.

**Figure 2 sensors-21-03075-f002:**
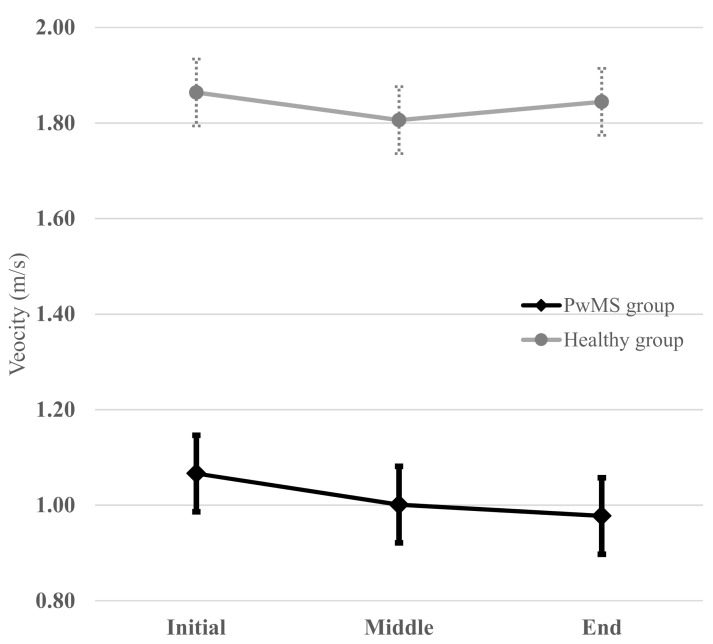
Averaged (i.e., mean of the test and retest values) walking velocity at each interval of the 6MWT for the PwMS and healthy group. PwMS had a “constant decline” pattern whereas healthy group had a “rebound” pattern (“V” pattern). The horizontal bars on each point represent the limits of standard deviation.

**Table 1 sensors-21-03075-t001:** Characteristics of the participants.

	PwMS (*n* = 45)	Healthy Group (*n* = 24)
Gender (female/male; N%)	26/19; 57.8/42.2	14/10; 58.3/41.7
Age (years)	51.4 (12.3)	51.3 (10.7)
Body mass (kg)	75.5 (18.3)	75.2 (13.7)
Height (m)	1.7 (0.08)	1.7 (0.08)
BMI (kg·m^−2^)	26.2 (5.5)	26.1 (3.9)
Tobacco exposition (y/n; N%)	11/34; 24.4/75.6	5/19; 20.8/79.2
Alcohol exposition (y/n; N%)	4/41; 8.9/91.1	0/24; 0/100
EDSS [4–6.5]	4.5 [4–5]	NA NA
EDSS classes (N; %)		
EDSS [4–5]	28 (62.2%)	
EDSS [5–6]	9 (20%)	
EDSS [6–7]	8 (17.8%)	
Disease duration (years)	17.4 (8.9)	NA
MS phenotype (N; %)		NA
RR	11 (24.4%)	
SP	20 (44.4%)	
PP	14 (31.1%)	

Abbreviations: BMI, body mass index; EDSS, Expanded Disability Status Scale; MS, Multiple Sclerosis; RR, Relapsing-remitting; SP, Secondary-Progressive; PP, Primary-Progressive; PwMS, Persons with Multiple Sclerosis; NA, Not Applicable. Values are means (SD) unless otherwise stated.

**Table 2 sensors-21-03075-t002:** Values of the walking variables and ICC for the test and retest and the three considered intervals of the 6-MWT of the PwMS group.

PwMS (*n* = 45)							
Variables	Test	6MWT Intervals			ICC [95% CI] ^c^		
		Initial	Middle	End	Initial	Middle	End
**Walking spatio-temporal ^a^**							
*Velocity (m/s)*	Test	1.03 (0.3)	0.97 (0.3)	0.96 (0.3)	0.864 (0.698–0.933)	0.879 (0.748–0.938)	0.890 (0.810–0.938)
	Retest	1.10 (0.3)	1.03 (0.3)	0.99 (0.3)			
*Cadence (step/min)*	Test	103.73 (13.5)	99.64 (14.2)	98.61 (14.4)	0.874 (0.679–0.942)	0.858 (0.696–0.929)	0.873 (0.754–0.933)
	Retest	107.42 (13.4)	103.18 (13.4)	101.57 (13.4)			
*Stride length (m)*	Test	1.18 (0.2)	1.15 (0.2)	1.16 (0.2)	0.907 (0.812–0.952)	0.919 (0.840–0.957)	0.909 (0.841–0.949)
	Retest	1.22 (0.2)	1.19 (0.2)	1.16 (0.2)			
*Stride width (m)*	Test	0.14 (0.04)	0.16 (0.04)	0.15 (0.04)	0.919 (0.858–0.955)	0.906 (0.835–0.947)	0.846 (0.737–0.912)
	Retest	0.14 (0.04)	0.15 (0.04)	0.15 (0.04)			
*Double support time (% GC)*	Test	32.83 (5.6)	34.36 (6.1)	34.75 (6.2)	0.912 (0.836–0.952)	0.889 (0.770–0.943)	0.913 (0.840–0.952)
	Retest	32.19 (5.6)	33.17 (5.8)	33.91 (6.1)			
**Borg scale (6–20) ^b^**	Test	10 [9–11]	12 [10–13]	13 [12–15]	0.322 (0.119–0.525)	0.279 (0.105–0.454)	0.376 (0.208–0.544)
	Retest	10 [9–10]	11 [10–13]	13 [12–14]			
**6MWT (m) ^a^**	Test	349 (100)			0.891 (0.791–0.942)		
	Retest	368 (102)					

**^a^** Mean (SD); **^b^** Median (First-Third quartile); **^c^** or weighted kappa (95% CI) for the reliability of the Borg scale in each interval of the 6MWT. Abbreviations: GC, gait cycle; ICC, intraclass correlation coefficient; 6MWT, 6-min walking test.

**Table 3 sensors-21-03075-t003:** Walking variable values and ICC for the test and retest and the three considered intervals of the 6MWT for the reference group.

Healthy Group (*n* = 24)							
Variables	Test	6MWT intervals			ICC [95% CI] ^c^		
		Initial	Middle	End	Initial	Middle	End
**Walking spatio-temporal ^a^**							
*Velocity (m/s)*	Test	1.81 (0.2)	1.78 (0.2)	1.80 (0.2)	0.691 (0.232–0.874)	0.775 (0.459–0.905)	0.682 (0.290–0.862)
	Retest	1.94 (0.3)	1.86 (0.3)	1.92 (0.3)			
*Cadence (step/min)*	Test	131.49 (8.3)	129.48 (8.2)	131.19 (10.1)	0.624 (0.079–0.849)	0.648 (0.180–0.852)	0.616 (0.260–0.818)
	Retest	137.62 (11.7)	134.84 (10.9)	136.43 (12.6)			
*Stride length (m)*	Test	1.65 (0.1)	1.64 (0.1)	1.64 (0.14)	0.900 (0.728–0.960)	0.946 (0.880–0.976)	0.808 (0.551–0.918)
	Retest	1.68 (0.1)	1.65 (0.1)	1.68 (0.14)			
*Stride width (m)*	Test	0.12 (0.02)	0.12 (0.02)	0.12 (0.02)	0.896 (0.772–0.954)	0.891 (0.767–0.951)	0.767 (0.534–0.892)
	Retest	0.13 (0.02)	0.12 (0.02)	0.12 (0.03)			
*Double support time (% GC)*	Test	23.12 (2.8)	23.93 (2.9)	23.56 (2.9)	0.788 (0.527–0.907)	0.788 (0.477–0.912)	0.569 (0.201–0.791)
	Retest	22.25 (2.8)	22.91 (2.8)	21.84 (3.7)			
**Borg scale (6–20) ^b^**	Test	9 [6–10]	9 [8–10]	9 [8–10]	0.788 (0.646–0.898)	0.734 (0.543–0.925)	0.786 (0.544–0.893)
	Retest	9 [6–9]	9 [9–10]	9 [9–10]			
**6MWT (m) ^a^**	Test	630 (67)			0.728 (0.350–0.886)		
	Retest	664 (88)					

**^a^** Mean (SD); **^b^** Median (First-Third quartile); **^c^** or weighted kappa (95% CI) for the reliability of the Borg scale in each interval of the 6MWT. Abbreviations: GC, gait cycle; ICC, intraclass correlation coefficient; 6-MWT, 6-min walking test.The Bland–Altman plots for each ST variable at each interval and distance in the 6MWT in both groups are illustrated in Appendix A. Excellent agreement (i.e., the mean of the differences close to 0 and tight LoA) was found between test and retest for all variables. More than 95% of observations were positioned within the LoA in all Bland–Altman plots.

## Data Availability

We certify that this is an original manuscript with no plagiarism or illegal data fabrication, that has not been published or submitted to another journal and that no party having a direct interest in the results of the research has or will confer a benefit on us or on any organization with which we are associated.

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
