# Peer review of "Evaluation of Prolonged Walking in Persons with Multiple Sclerosis: Reliability of the Spatio-Temporal Walking Variables during the 6-Minute Walk Test"

_sensors, 2021, doi:10.3390/s21093075_

Round 1
Reviewer 1 Report
This manuscript investigates the reliability of several spatiotemporal variables (and perceived exertion) during the 6MWT in persons with MS. The authors should be commended for their efforts and for producing a manuscript that was well written and easy to follow. The manuscript would be of interest to those using the 6MWT in persons with MS. I have no major concerns with the manuscript – I have provided a number of suggestions below to help improve clarity of the manuscript.
Title: to be succinct, I suggest removing ‘the measurement of the’ as the title would be the same without it.
Abstract, line 15: what aspect of the 6MWT have previous studies shown good reliability? Is it just for the distance walked or the average speed? I suggest stating this as you then go on to analyse spatio-temporal variables specifically.
Introduction, line 33: can you define what age you mean when you say ‘young adults’?
Introduction, line 35: Minor point but the abbreviation PwMS for the title and abstract is ‘patients with MS’ but in the Introduction, you use ‘persons with MS’. I suggest being consistent – perhaps persons with MS is a better term.
Introduction, line 36 (and throughout): you do not need ‘the’ before ‘PwMS’ – so delete ‘the’ here and any other time it appears before PwMS.
Introduction, line 51: ‘lower ecological properties’ – lower compared to what?
Introduction, line 57: suggest removing the word ‘currently’ as the 6MWT has been evaluated for a long time – so maybe rephrase to ‘have been evaluated for use…’.
Introduction, line 68: perhaps change ‘in one of them’ to something like ‘in one study in X population’ to indicate what other condition they studied.
Introduction, line 81-82: when you say ‘little is known about the reliability of the ST variables measures during the 6MWT’ – is that just in PwMS or is that in any population? I also suggest removing ‘measures’ from this part of the text.
Introduction, line 86: suggest using ‘reliability’ instead of ‘reproducibility’ here for consistency.
Methods, line 95: were the walking assessments done prior to the intervention? If so, please state here.
Methods, line 97: can you provide a reference for the McDonald criteria?
Methods, line 104: suggest rewording ‘constituted’ with ‘recruited’ or something similar.
Methods, line 104-106: were the healthy group specifically recruited based on these characteristics or did it just turn out that there were no differences between groups in these characteristics? This is an important detail about how you recruited the healthy group.
Methods, line 108: this sentence is not clear and should be reworded.
Methods, line 138: please reword ‘these walking analyze respect the number..’
Methods: was any warm-up or practice provided prior to the trials? And what instructions were provided to the participants?
Table 1: suggest changing ‘size’ to ‘height’ or ‘body height’. Also, how was tobacco and alcohol exposition recorded? This should be detailed in the methods about the questions or surveys asked to define this as ‘yes’ (e.g., was it any tobacco/alcohol use or was it regular use or something else).
Results, line 195: your definition of excellent reliability was above 0.90, but the reliability of distance for the MS group was 0.891 so this should be good.
Results, line 198: Table 2 has one of the ICC values as 0.846 which is lower than your reported lower range of ICC values here in this line of text.
Table 2: the Borg scale is for a weighted kappa, but in this table, it looks like it is an ICC value based on the headings. Maybe put a footnote in about this? Also, should the symbol next to the 6WMT distance be ‘a’ instead of ‘b’ as it looks like it is presented as a mean value not a median value.
Figure 2: can you describe what the bars indicate just to be clear to the reader – is it standard deviations? Also, I suggest changing to use decimal points for your axis labels (i.e., 1.2 instead of 1,2).
Results: how many times in the minute intervals did participants cross the GAITRite? In the methods you said they had to have at least one crossing, but did some people have multiple crossings? And was that then consistent for the second testing session?
Discussion, line 257: Are you saying people with more disability had better reliability? Did you test this? Or sorry, when you say ‘this study’ are you referring to your manuscript or the previous study – this might be where my confusion is.
Discussion, lines 289-292: I assume this text should be deleted – it appears to be instructions to the authors.
Reviewer 2 Report
The authors examine the common 6-min Walk Test, and examined the gait parameters of MS patients at the start/middle/end of the test, while also collecting their Borg extension. Each participant is captured twice, alongside with a set of healthy baseline participants. The 6MWT was shown to provide consistent result across different collection time (i.e. test-retest validated), and time point (i.e. different parts of the 6MWT) over both healthy and patient populations
While the proposed paper is clear and well written, I have some comments and questions below
- Please be more explicit at the novelty of the paper. It seems like [19], like the proposed paper, also examine 6MWT in a piecewise and objective fashion. Is the proposed paper's novelty that it recollected the same participant twice at a different time point? Also, ST gait parameters also have been examined by [21-22]. Is the novelty here that both prior papers are combined into this paper? Given the popularity of the 6MWT and the large number of papers that examine its various aspects, it may help to provide a simple table or diagraph that emphasize the novelty
- The paper focuses on t=[0, 1 min], [2.5, 3.5 min] and [5, 6 min]. Do the participants generally complete one lap of the 24-m track per minute? Or do they generally complete more laps, and thus some of the collected data were discarded? Looking at Table 2, it seems like that patients in general complete much more laps than is reported. I am assuming that these time intervals were chosen to match prior work, but there potentially could be more insight obtained if more data were examined. By your own introduction, continuous and subject data is difficult to come by. Or was finer grain data analyzed, but it did not show anything significantly different than what is reported in Table 2, etc.?
- I see now on line 152 that each of the ST over the time interview was averaged together, as a response to my Q2. I am still wondering about not averaging together the data and break down the 6MWT to smaller intervals
- This is not a question, but I am impressed by the healthy group population, that your team was able to find participants of similar age, weight, height, as well as similar gender and tobacco/alcohol distribution.
- In Table 1, should the square bracket in EDSS be the other way? i.e. [4-5]?
- Would you expect the ICC to have been higher for the patient population if the time between rest/retest were shorter?
- Can you comment on why you think the ICC for healthy participants are lower? I would've expected the healthy population to be fairly consistent
- Fig 2 is quite interesting. Are the participants (both healthy and patients) informed of how much time has elapsed? Does the "anticipation" that the experiment is ending explain the uptick for the healthy participants "V" pattern? Also, do these velocities differ significantly from each other? They seem to be fairly close and well within each other's standard deviations. While the logic that the MS patients sees the 6MWT as a maximum stress test vs the healthy participant does not is a sound one and makes a lot of sense to me, the variance in the data might be too low to show this
- In the discussion section, while it is evident that test-retest reliability to be high for patients, and agree with various prior studies, it seems less obvious for healthy walkers. Have there been prior experiments that looked at 6MWT retest reliability for healthy participants?
- Line 289 seems to be part of a template and should be removed
Reviewer 3 Report
This is a well written manuscript. The experimental details are sufficient for replication and the statistical analysis is thorough and attentive to detail. Overall, the manuscript is quite refined and technically sound.
A lot of previous work exists on the 6MWT, and while the authors appear to have cited most of the major works, it is somewhat unclear what is truly novel and what is confirmatory. The authors could include a more clear statement of how their study contributes new information that isn't already published.
Specific Comments
Abstract, Line 14: Contemporary language is "persons" or "people" and not "patients". PwMS still works, so no need to change.
Pg 3, line 104: From where was the healthy group recruited? From the general community? Local university? Hospital staff?
Pg 10, line 243-258: Somewhat rambling and might be condensed.
Pg 10-11, lines 289-292: These sentences do not appear to be intended for the manuscript.
Round 2
Reviewer 2 Report
Thank you for address my concerns. I have no additional questions.